# Dynamic of the Soil Microbiota in Short-Term Crop Rotation

**DOI:** 10.3390/life13020400

**Published:** 2023-02-01

**Authors:** Aleksei O. Zverev, Oksana N. Kurchak, Olga V. Orlova, Olga P. Onishchuk, Arina A. Kichko, Aleksander V. Eregin, Aleksey N. Naliukhin, Aleksandr G. Pinaev, Evgeny E. Andronov

**Affiliations:** 1All-Russian Research Institute for Agricultural Microbiology (ARRIAM), 3 Podbelsky Chaussee, St. Petersburg 196608, Russia; 2V.V. Dokuchaev Soil Science Institute, 7 Building 2, Pyzhyovskiy Lane, Moscow 119017, Russia; 3N.V. Vereshchagin Vologda State Dairy Farming Academy, 2 Shmidta Street, Molochnoye, Vologda 160555, Russia; 4Timiryazev Moscow Agricultural Academy, Russian State Agrarian University, 6 Pryanishnikova, Moscow 127434, Russia

**Keywords:** crop rotation, 16S rDNA gene sequencing, soil microbiota, soil exhaustion, soil sequencing

## Abstract

Crop rotation is one of the oldest and most effective methods of restoring soil fertility, which declines when the same plant is grown repeatedly. One of the reasons for a reduction in fertility is the accumulation of pathogenic and unfavorable microbiota. The modern crop rotation schemes (a set of plant species and their order in the crop rotation) are highly effective but are designed without considering soil microbiota dynamics. The main goal of this study was to perform a short-term experiment with multiple plant combinations to access the microbiological effects of crop rotation. It could be useful for the design of long-term crop rotation schemes that take the microbiological effects of the crop rotation into account. For the analysis, five plants (legumes: vetch, clover, and cereals: oats, wheat, and barley) were used. These five plants were separately grown in pots with soil. After the first phase of vegetation, the plants were removed from the soil and a new crop was planted. Soil samples from all 25 possible combinations of primary and secondary crops were investigated using v4-16S rDNA gene sequencing. It was shown that the short-term experiments (up to 40 days of growing) are effective enough to find microbial shifts in bulk soil from different plants. Both primary and secondary cultures are significant factors for the microbial composition of microbial soil communities. Changes are the most significant in the microbial communities of vetch soils, especially in the case of vetch monoculture. Growing clover also leads to changes in microbiota, especially according to beta-diversity. Data obtained can be used to develop new crop rotation schemes that take into account the microbiological effects of various crops.

## 1. Introduction

It is known that the continuous growth of a single crop leads to soil exhaustion and fatigue. This effect includes plant growth retardation and low productivity, pest and disease accumulation, and soil structure and nutrition degradation [1]. Crop rotation is one of the most efficient ways to prevent soil fatigue and increase the productivity of crops. Crop rotation improves soil physical structure and aggregation, increases nutritional components and diversity of soil microbiota and associated beneficial microbes, and controls soilborne pathogens by disrupting their natural life cycle [2].

The mechanism of these beneficial changes is based on plant exudation and, in the case of cover crops (i.e., crops that are not harvested and plowed into the soil), the biomass of plants. The main components of plant exudates are carbohydrates, amino acids, and organic acids [3]. This carbon intake leads to nutrition enrichment of the soil, which is the most widespread and well-known effect of crop rotation. The meta-analysis of 122 reports reveals that adding one or more crops in rotation to a monoculture increased total soil C by 3.6% and total N by 5.3%. Moreover, when rotations include a cover crop, total C increases by 8.5% and total N by 12.8% [4]. In the analysis of 11 maize crop rotation schemes in North America, Bowels and co-authors showed that more diverse crop rotation leads to (i) yield growth acceleration, (ii) increased yields under harsh conditions, and (iii) yield benefits in different growing conditions [5].

According to different studies, microbial diversity, enzymatic activities, and soil respiration are generally higher in crop rotation systems than in monoculture or fallow [2]. Exact changes in microbial communities are highly specific and are determined by a range of factors, including plant species, nutritional composition, and soil type [6]. For example, in comparison of tomato crop rotation schemes (cabbage/tomato, kidney bean/tomato, and celery/tomato), the celery/tomato scheme demonstrated the best performance [1]. Another trial with a partial addition of grassland in the crop scheme (no grassland; 50% grassland; 75% grassland) found that forage production was higher in all crop rotation schemes with grassland, but there were no benefits for 75% of grassland sites [7]. Despite all the positive information, not all crop rotation schemes are better than monoculture cultivation. For example, Koyama and co-authors showed a decrease in yield in some crop combinations (canola after alfalfa) [8]. A more wide-ranging meta-analysis also shows that some crop diversification strategies are more effective than others [9].

There is no unique formula for the best crop rotation. Alongside the whole range of factors (soil, climate, economic circumstances), the main concerns are: (i) greater diversity of crops in time and space; (ii) alternation of crops with different rooting depths; (iii) restoration of soil fertility; (iv) increasing the innate capacity of crops and soils to suppress weeds, pests, and diseases and avoid soil fatigue; and (v) preventing soil erosion and drought [10]. Most of the current crop rotation schemes were constructed according to these recommendations.

Today, in addition to classic nitrification and chemical analysis, the researchers can use another type of analysis—NGS sequencing. Multiple studies of crop rotations reveal changes in either bacterial or fungal rhizosphere communities [1,2,11]. More hidden and complicated changes in the bulk soil microbiota were also revealed [12].

Knowing the positive effects of different plants in rotation and how the microbial community responds, the question arises: can we use NGS sequencing to predict an even more effective crop rotation scheme? What are the predictors of a good scheme?

Kaplan and co-authors evaluate the relationship between phylogenetic distance and crop performance. In their estimation of tomato production within a rotation with 35 different crops and weeds, they do not find an effect of phylogenetic distance on crop performance [13], although there were significant changes in microbial communities.

In search of the most effective crop rotation scheme, we have to test multiple combinations of crops and plants over a long period of time. This takes a lot of time and effort. For estimating the most effective plant combinations, prior short sub-experiments are reasonable. Direct evaluation of crop productivity in this short experiment is difficult. Instead, the changes in the microbial community of the soil can be used as a strong marker of changes in soil nutrition. Research on these changes is also novel due to the lack of information about microbial changes in soils under different crop rotation schemes.

For the analysis, five widespread agricultural plants (legumes: vetch, clover, and cereals: oats, wheat, and barley) were used. Initially, these five plants were grown in untreated soil. After this, the soil from every plant was mixed and again seeded by the same five plants. After a short vegetation period, the soil was sampled and stored. As a result, there were 25 samples with all combinations of primary and secondary cultures. The microbial diversity of soil samples was estimated using V4 16S rDNA gene sequencing. We calculate the crop rotation effect based on the impact of plants on the entire soil microbial community and forecast good crop rotation combinations for a large-scale crop rotation experiment.

Despite the advantages of 16S rDNA gene sequencing, this method has its own limitations. In the future, whole genome sequencing or amplicone sequencing of functional genes will be preferable.

## 2. Materials and Methods

Both the soil and the seeds for the experiment come from the Vologda region (western Russia), which is one of the main crop harvesting centers. The sod-podzolic soil was mixed, sieved, and air-dried; enriched with NPK fertilization mix (0.071 g NH_4_NO_3_, 0.02 g K_2_HPO_4_, and 0.036 g KH_2_PO_4_ per pot); watered to 50% of its maximal moisture capacity; and placed in 10 cm × 100 cm × 15 cm pots. Plant seeds (from real crop rotation schemes) were disinfested in H_2_SO_4_ for 5–8 min.

For the analysis, five widespread agricultural plants (legumes: vetch, clover, and cereals: oats, wheat, and barley) were used. These 5 plants were grown in 29 days (phase I). After sampling, the soil from every plant was mixed individually, generating five unique soils. Then, every unique soil was seeded again by the same 5 plants (6 repeats for every plant). After a 22-day vegetation period (Phase II), the soil was sampled again. As a result, there were 25 samples with combinations of five primary and five secondary cultures. For reference samples (pots without seeds), the same treatment was used.

After every phase of the experiment, sampling was made: plants, roots, and debris were removed from all soil in the pot, the bulk soil was mixed, and 2 g was sampled for the microbial analysis.

Total soil DNA was isolated using the MN NucleoSpin Soil Kit (Macherey-Nagel, Düren, Germany) using a Precellus 24 homogenizer (Bertin, Rockville, MD, USA). The quality control of the isolation was carried out by PCR and agarose gel electrophoresis. DNA sequencing using the V4 variable region of the 16S rRNA gene was performed on an Illumina MiSeq sequencer (Illumina Inc., San Diego, CA, USA), using the primers 515f (GTGCCAGCMGCCGCGGTAA) and 806r (GGACTACVSGGGTATCTAAT) [14]. Raw reads are available on SRA (accession number PRJNA891634), or by a direct link: https://www.ncbi.nlm.nih.gov/sra/PRJNA891634 (accessed on 14 November 2022).

The general processing of sequences was carried out in R 4.2.0 (R Foundation for Statistical Computing, Vienna, Austria), using the dada2 (v. 1.14.1) [15] and phyloseq (v. 1.30.0) [16] packages, according to the authors’ recommendations. 16S rDNA gene sequences were processed according to the DADA2 pipeline. Sequences were trimmed for length (minimum 220 bp for forward reads and 180 bp for reverse reads) and quality (no N; maximum error rates maxEE of 2 for both forward and reverse reads). ASVs were determined according to the DADA2 algorithm, and chimera ASVs were removed by the “consensus” method. Taxonomic annotation was performed by the naive Bayesian classifier (provided in the DADA2 package, default settings) with the SILVA 138 database [17] as the training set, and chloroplast and mitochondrial reads were eliminated. The final sequence yield was from 10,779 to 26,577 reads per sample. The alpha-diversity calculation was estimated using rarefaction to minimal depth (10,779).

The main alpha- and beta-diversity indices were calculated using ASVs and their abundances using the phyloseq package. PERMANOVA was performed using the vegan package [18].

## 3. Results

### 3.1. Phase I

The richness of ASVs in soil communities is presented in Figure 1A. The alpha-diversity of the fallow soil microbiota is the lowest in the dataset (mean 2705 ASVs), while microbial communities after the setting of soil legacies are more diverse. According to a one-way ANOVA, the primary culture is a significant factor (see Appendix A). Microbial communities after barley are significantly less diverse than those from other crops and do not differ from fallow samples. The Simpson index (according to ANOVA) reveals no differences between all samples.

According, to beta-diversity indices, microbial communities are similar. According to PCoA ordination, the values of explained variance are less than 5% (Figure 1B). Despite this, PERMANOVA reveals primary culture as a significant factor (see Appendix A).

### 3.2. Phase II

The values of microbial richness are similar to those in Phase I (Figure 2). According to a two-way ANOVA (fallow samples were excluded), primary culture and the combination of primary and secondary cultures were significant factors for Observed ASV index (see Appendix A).

Beta-diversity was calculated using Bray-Curtis distances. According to PERMANOVA, both primary and secondary cultures and their combination have a significant effect (see Appendix A). PCoA was used to ordinate the distances (Figure 3). As for phase I, the values of explained variance by the first and second axes are low, so for the clearance of the effect we also use Ward’s diagram (Figure 4).

## 4. Discussion

In phase I, there were moderate changes in the soil microbial community. According to the richness of ASVs and ANOVA, microbial diversity in all crop soils (except barley) is about 2920, while microbial diversity in fallow soil and bulk soil after barley is significantly less—2780. According to Bray-Curtis distances, microbial communities in all crop soils are different, but this difference is small. Therefore, it is reasonable to suggest that microbial communities in bulk soils of different crops are different, but the scale of their changes is comparable to intergroup variances. This result is reasonable due to the short period of cultivation. According to reports on the rhizosphere analysis of sorghum, the effect of plant genotype is observed only after 35 days [19].

Despite moderate changes in phase I, the primary culture is significant as a baseline for the microbial communities of soils in phase II. According to variance analysis, primary culture, and the combination of primary and secondary culture (but not the secondary culture itself) were significant factors for ASVs’ richness. All factors (primary and secondary cultures and their combinations) were important in the PERMANOVA analysis of Bray-Curtis distances.

The strength of both factors (primary and secondary cultures) is ambiguous. The variance analysis allows us to determine primary culture as a significant factor for both alpha (richness in phase II) and beta (Bray distances) diversities. But the percent of explained variability for the PCoA plot is very low. This effect can be explained both by a short period of experimentation (20–30 days for each phase) and bulk soil sampling. As it was mentioned earlier, the main effect of crop rotation is connected with the rhizosphere microbial community, not the bulk soil one. Changes in the bulk soil microbial community have to be moderate. For example, in the long-term analysis of field grassland crop rotation systems, the changes in microbial soil diversity were significant but relatively weak [7]. In the comparison of wheat monoculture and pea-wheat crop rotation (two years of experiment), there was no difference in alpha-diversity and a moderate difference in beta-diversity and taxonomic composition [2]. Therefore, the changes in the microbial community of the bulk soil during crop rotation are moderate, even in long-term experiments. Both studies found significant variation in fungal communities, particularly arbuscular mycorrhizal fungal (AMF) communities [2,7].

Even in this relatively short experiment, it is possible to find the significant influence of different crop rotation schemes on the microbial composition of soils. The influence of cultures on soil microbiota is variable. Growing legume plants as the second culture results in a higher beta-distance between samples than for the cereal ones (Figure 4). Vetch has the greatest impact on microbial composition in this legume group. In the dendrogram of Bray-Curtis distances (Figure 3), all vetch samples (both for primary and secondary culture) are located in separate clusters. One of the possible reasons is the alteration of soil nitrogen metabolism, even in non-cover crop rotation. This mechanism was previously shown as a beneficial element in a rice-based crop rotation system in China [20]. In contrast, cereal plants—the barley and the wheat—have no significant effect on soil composition. Presumably, the oat has the strongest effect (Figure 3), but this suggestion needs additional research.

Another point of the experiment was connected with the effects of continuous growth of the same plant. Due to the short period of growth, the exhaustion of the soil cannot be revealed. Furthermore, the microbial communities of vetch after vetch soils differ significantly from others, according to Bray-Curtis distances. These changes do not include any specific taxonomic shifts or changes in alpha-diversity. The soil microbiota of clover after clover showed a similar but less pronounced effect. Both shifts can be connected with the alteration of nitrogen metabolism.

Even a short period of plant growth allows us to determine changes in the microbial composition of soils. These changes are moderate and masked by high intergroup variance, but there is a clear effect of using legumes in crop rotation schemes. For clarification of this effect, it is required to set up the following experiments: (i) comparison of legumes, legumes and cereals, and cereal crop rotation schemes; (ii) analysis of both the microbial and fungal communities.

## 5. Conclusions

Short-term experiments (20 to 40 days) allow for the discovery of microbial shifts even in bulk soil from different plants. In the first phase, the microbial communities of soils from all plants except barley are close. The alpha diversity of microbial communities in soils from any plant, except barley, is higher than the one from reference fallow soil. For the whole experiment, both primary and secondary cultures are significant factors for the microbial composition of microbial soil communities. Despite this, the strength of those factors is limited. In our study, vetch showed the strongest effect on the microbial composition of soil, especially continuously growing vetch. According to beta-diversity, growing clover also leads to changes in soil microbiota. The influence of other cultures is moderate and, as a rule, does not exceed the intergroup variation.

## Figures and Tables

**Figure 1 life-13-00400-f001:**
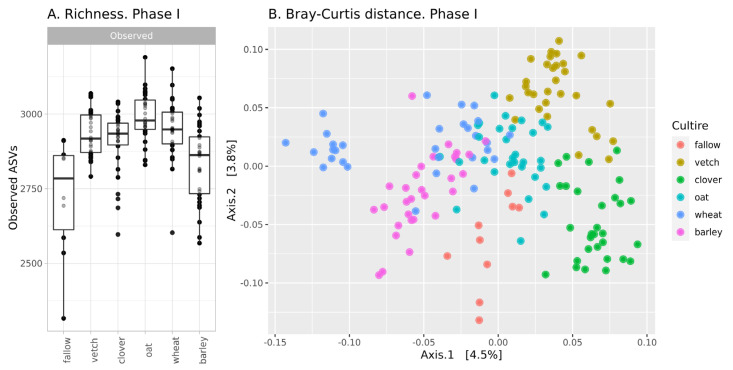
Changes in the microbial community of soils in the first phase of the experiment (**A**): richness of samples; (**B**): Bray-Curtis distance between.

**Figure 2 life-13-00400-f002:**
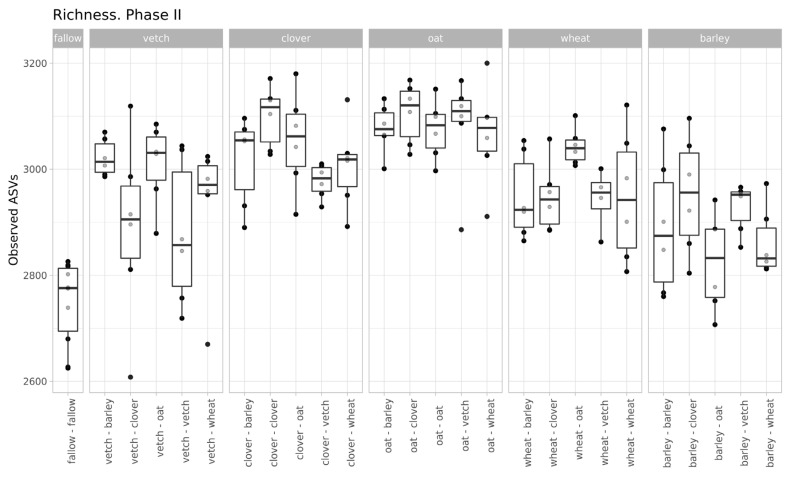
Richness of samples (according to the Observed ASVs Index, grouped by primary culture).

**Figure 3 life-13-00400-f003:**
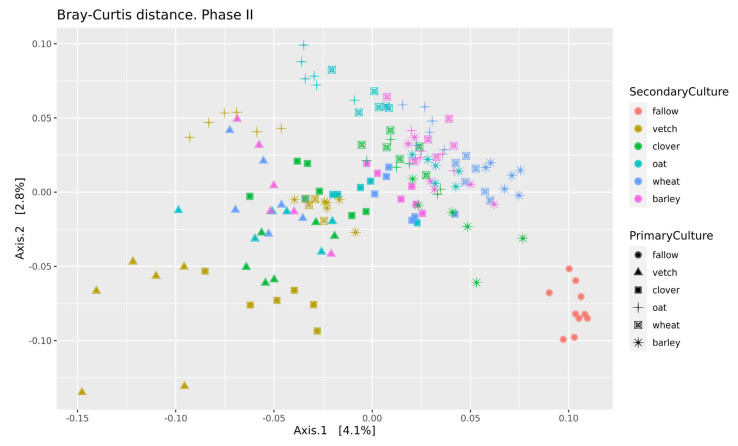
PCoA of Bray-Curtiss distances in soil microbial communities. Primary Culture—plant type in phase I; Secondary Culture—plant type in phase II.

**Figure 4 life-13-00400-f004:**
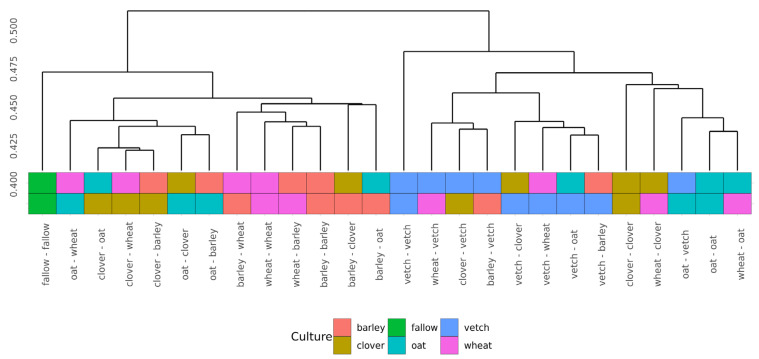
Cluster dendrogram (Ward’s clustering) of Bray-Curtiss distances in microbial communities of soils.

## Data Availability

Raw reads are available on SRA (accession number PRJNA891634) or by a direct link: https://www.ncbi.nlm.nih.gov/sra/PRJNA891634 (accessed on 14 November 2022).

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
