# Peer review of "Dynamic of the Soil Microbiota in Short-Term Crop Rotation"

_life, 2023, doi:10.3390/life13020400_

Round 1

Reviewer 1 Report

The authors present the development of an interesting express-test determining suitable crop rotations based on soil microbiome dynamics.

The usage of tense varies within the manuscript and should be adjusted to a standard. Further, I recommend an extensive editing of English language and style.

The figures need more comprehensive captions in order to be interpretable.

Please comment on how many replicates per sample you evaluated and on the innergroup variance. Is the innergroup variance higher than the intergroup variance?

Please comment on the disadvantages of microbial composition determination via 16S rRNA gene sequencing - changes on soil microbial community can be on species level, which can only be resolved with metagenome sequencing.

How was the soil sampled? 1g? Root-influenced or pure bulk soil?

L. 9 ...declining when growing the same plant repeatedly.

L.9 One of the mechanisms for reducing fertility - reducing fertility is not something we want to achieve, so rather than speaking of mechanisms better "One of the reasons for a reduction of fertility is..."

L. 16 five

L. 22 three times vetch?

L. 24 a

L. 32 system

L. 44 Addition of grasslands? Do you mean green manure as part of the crop rotation? Or strips of grass planted in between?

L. 47 another other

L. 48 According to different studies, microbial diversity, enzymatic activities, as well as

L. 49 that than

L. 51 including

L. 52 In different experiments, crop rotation increased...

L.55 the crop rotation effect

L. 107 DNA was sequencing

L. 137 & 138 Bray-Curtis

L. 153 why should the method ordinate "correctly"? Also, you can't state "Therefore, here the Ward's diagramm" in a scientific manuscript.

L. 193 studies

L. 208 Soil exhaustion by means of lower microbial diversity or by means of soil nutrition ?

LL. 2019-222 Is this the text of a document on how to write a scientific paper?

L. 230 In our study, vetch showed the strongest effect

Please comment in the conclusion, how your proposed express-test can be methodologically refined based on your study. E.g. using whole metagenome sequencing, longer growing periods, sampling of root-influenced soil or rhizosphere soil rather than bulk soil,...

Further, a scheme of the express-test setup would be of great value for the manuscript. Visualising  the five different crops and the first phase/second phase combination would ease to follow the setup.

Author Response

The authors present the development of an interesting express-test determining suitable crop rotations based on soil microbiome dynamics. The usage of tense varies within the manuscript and should be adjusted to a standard. Further, I recommend an extensive editing of English language and style.

Authors: we made our best, alongside it we will use MDPI proofreading system

The figures need more comprehensive captions in order to be interpretable.

Authors: figures were redrawn, captions were extended

Please comment on how many replicates per sample you evaluated and on the innergroup variance. Is the innergroup variance higher than the intergroup variance?

Authors: Materials and Methods section was rewritten, information about repeats is in L:107-113

Please comment on the disadvantages of microbial composition determination via 16S rRNA gene sequencing - changes on soil microbial community can be on species level, which can only be resolved with metagenome sequencing.

Authors: the amendments were made

How was the soil sampled? 1g? Root-influenced or pure bulk soil?

Authors: Materials and Methods section was rewritten, information about sampling is in L:114-116

L. 9 ...declining when growing the same plant repeatedly.

Authors: the sentence was corrected

L.9 One of the mechanisms for reducing fertility - reducing fertility is not something we want to achieve, so rather than speaking of mechanisms better "One of the reasons for a reduction of fertility is…"

Authors: the sentence was corrected

L. 16 five

Authors: the sentence was corrected

L. 22 three times vetch?

Authors: Repetative groving of vetch, both first and second culture. The sentence was corrected

L. 24 a

Authors: the sentence was corrected

L. 32 system

Authors: the sentence was corrected

L. 44 Addition of grasslands? Do you mean green manure as part of the crop rotation? Or strips of grass planted in between?

Authors: we suggest using grassland in crop rotation scheme. The sentence was ambigious, it was corrected

L. 47 another other

Authors: the sentence was corrected

L. 48 According to different studies, microbial diversity, enzymatic activities, as well as

Authors: the sentence was corrected

L. 49 that than

Authors: the sentence was corrected

L. 51 including

Authors: the sentence was corrected

L. 52 In different experiments, crop rotation increased…

Authors: the sentence was corrected

L.55 the crop rotation effect

Authors: the sentence was corrected

L. 107 DNA was sequencing

Authors: the sentence was corrected

L. 137 & 138 Bray-Curtis

Authors: the term was corrected in the whole manuscript

L. 153 why should the method ordinate "correctly"? Also, you can't state "Therefore, here the Ward's diagramm" in a scientific manuscript.

Authors: the sentense was messy, the whole paragraph was rewritten

L. 193 studies

Authors: the sentence was corrected

L. 208 Soil exhaustion by means of lower microbial diversity or by means of soil nutrition ?

Authors: the sentence was clarified

LL. 2019-222 Is this the text of a document on how to write a scientific paper?

Authors: Yes, it’s a mistake. The text was cleared

L. 230 In our study, vetch showed the strongest effect

Authors: the sentence was clarified

Please comment in the conclusion, how your proposed express-test can be methodologically refined based on your study. E.g. using whole metagenome sequencing, longer growing periods, sampling of root-influenced soil or rhizosphere soil rather than bulk soil,…

Authors: we leave the idea of express-test, reframe the research and now present it as a first approach for the new type of crop rotation - using information about changes in microbiota in wide range of plant combination. We rewritten the conclusion.

Further, a scheme of the express-test setup would be of great value for the manuscript. Visualising  the five different crops and the first phase/second phase combination would ease to follow the setup.

Authors: we completely rewritten the M&M section, in particular, the design part. We are ready to provide graphical scheme of the experiment, if nessesary.

Reviewer 2 Report

The authors did some interesting experiment to study the effects of two-round rotation of plants on soil microbial communities. Efficient statistical analysis have been done and displayed for the microbial community study. However, there are some obvious flaws in the experimental design, such as short experimental periods (40 days), collection of soil away from plants rhizosphere and studying only bacteria while neglecting fungal communities, etc. All of these may cause ambiguous conclusions in the study. English also needs to be improved. 

Minor comments:

Line 126. Mention clearly in the methods what are these samples and how you collect them (e.g., sampling depth, etc)

Line 131. According to...

Line 184-187. Needs to be clear in methods about the type of soil collected (rhizosphere or bulk)

Author Response

The authors did some interesting experiment to study the effects of two-round rotation of plants on soil microbial communities. Efficient statistical analysis have been done and displayed for the microbial community study. However, there are some obvious flaws in the experimental design, such as short experimental periods (40 days), collection of soil away from plants rhizosphere and studying only bacteria while neglecting fungal communities, etc. All of these may cause ambiguous conclusions in the study. English also needs to be improved. 

Authors: The main studies, indeed, use long-term experiment and rhizosphere sampling. However, here we try to investigate, is it possible to find the differences in such relaxed conditions. In most cases the rhisosphere soil is used – but these changes are connected to root surface. Changes in bulk soil are more moderate, but also more valuable. We also are going to perform the long-term experiment, using the data from this one

Line 126. Mention clearly in the methods what are these samples and how you collect them (e.g., sampling depth, etc)

Authors: Materials and methods section was rewritten

Line 131. According to…

Authors:  the sentence was corrected

Line 184-187. Needs to be clear in methods about the type of soil collected (rhizosphere or bulk)

Authors: Materials and methods section was rewritten

Reviewer 3 Report

This study developed an "express method" for accessing the microbiological effects of crop rotation. To assess that, five agricultural plants (two legumes and three cereals) were tested in a greenhouse experiment. The authors used amplicon sequencing to describe microbial communities from 25 different combinations.

The study is interesting and quite relevant. The sampling design, methodology, and statistical analyses seem appropriate and properly applied, and therefore I believe the results are of interest to scientists working on the subject.

Three main points to highlight: (a) in line 69, the authors state that most crop rotation schemes are done empirically, but no references were presented. Perhaps, Bowles et al. (2020) is a good citation for that discussion. Is it possible to add some more references or statistics about it? (b) Even that, I do not see the manuscript's main idea as an “express test”, since there is no methodology described that can be guided as a test to further crop rotation decision experiments. What is described is experiments showing that the previous crop matters in terms of defining the community for the next crop. (c) The beta diversity by ordination analysis must be tested with no rarefied data. I suggest that the ordination plot be presented in the manuscript even if the results are not well appreciated by the author.

Following, I picked up some specific points that I suggest to be carefully improved.

Minor comments

L89 – 16S rRNA gene sequencing (not 16S rDNA sequencing) is more appropriate to define the amplification of this marker gene. Please apply this also for the remaining text (e.g., L114). Also, check the standardization proposal by Marchesi and Ravel (2015).

L93 – To better localize the reader, it would be good to add the country where the Vologda region is situated (Vologda region, Russian). Please, include it.

L97 – Seed sterilization is not possible because the endophytes are still inside. The term “disinfestation” is more appropriate. Please, change it.

L99 – What does untreated soil mean? Is it from the same region? Please clarify. Also, it would be easier for the reader to understand if a sentence is added highlighting that “the soil from every plant was mixed individually generating five unique soils. Then, every soil was seeded …).

L122 – Please, change “beta-metrics” to “beta diversity”. It is the most appropriate term to be used. Also, include what was used to calculate the Bray-Curtis distance, the ASVs or any combined taxa level.

L149 and L156 – Please, provide the table with all statistical results.

L136, L138, L153 – Bray-Curtis is misspelled. Please check the entire manuscript and figures, including legends.

L137 – It is said that PCoA and NMDS could not clarify the diversity. It might be an effect caused by normalization (rarefaction). For alpha diversity, rarefaction is a common practice. But for beta diversity, the best approach would be using a square root normalization. When rarefying data, many species in low abundance are rarefied. They are important in the differences observed in the overall beta diversity. Please, look at McMurdie et al. 2014.

L146 – After Phase I, no need to add “one”.

L155 - Please, change “median distance” to “Bray-Curtis distance”.

L177 – Observed is a metric to measure the richness of species. Please, change “Observed” to “richness”. Please apply this also for the remaining text.

L216- L222 – Please check the text content here. It seems that has something wrong, or it doesn’t belong to your manuscript. Remove it or rewrite it in a meaningful way.

L224-L225 – Please, check your conclusion. It seems contradictory to what you wrote on L172-L173 or your Bray-Curtis results. 

FIGURE 1 – In Fig. 1A, change the Y axis to “Species richness” (instead of “Observed index”). Also, in the figure title, it should be “A. Richness” (instead of “A. Alpha-diversity). In Fig. 1B, the title should be “B. Bray-Curtis distance” (instead of Average …). In Fig. 1C, the title should be “C. Differential relative abundance”. What does the NA mean? Also, italic genera should be italicized. 

FIGURE 4 – This figure is not informative for your study. I suggest remove it from the manuscript.

Suggested references

Marchesi JR, Ravel J (2015) The vocabulary of microbiome research: a proposal. Microbiome 3:31.

Bowles et al. (2020) Long-term evidence shows that crop-rotation diversification increases agricultural resilience to adverse growing conditions in North America. One Earth 2: 284-293.

McMurdie PJ, Holmes S (2014) Waste Not, Want not: Why rarefying microbiome data is inadmissible. PLOS Computational Biology 10(4): e1003531

Author Response

This study developed an "express method" for accessing the microbiological effects of crop rotation. To assess that, five agricultural plants (two legumes and three cereals) were tested in a greenhouse experiment. The authors used amplicon sequencing to describe microbial communities from 25 different combinations. The study is interesting and quite relevant. The sampling design, methodology, and statistical analyses seem appropriate and properly applied, and therefore I believe the results are of interest to scientists working on the subject. Three main points to highlight: (a) in line 69, the authors state that most crop rotation schemes are done empirically, but no references were presented. Perhaps, Bowles et al. (2020) is a good citation for that discussion. Is it possible to add some more references or statistics about it? (b) Even that, I do not see the manuscript's main idea as an “express test”, since there is no methodology described that can be guided as a test to further crop rotation decision experiments. What is described is experiments showing that the previous crop matters in terms of defining the community for the next crop. (c) The beta diversity by ordination analysis must be tested with no rarefied data. I suggest that the ordination plot be presented in the manuscript even if the results are not well appreciated by the author.

Authors: thanks for so relevant and comprehensive review. Using your highlights, we rewritten several parts of the manuscript. a) we included part about construction of crop rotation. Our point was that crop rotations were created without information about microbial diversity. Sure enough , they were not a random sequence of plants “empirical”. The amendmends was added in the introduction. b) you are right, after a discussion we desided to present the research not as a test system, but as a first approach to understand the dynamic of the soil microbiota in short-term crop rotation. c) the rarefied data was used for alpha-diversity only. This was not clear from materials and menhods, indeed. The section was rewritten completely. Also we use PCoA plots instead heatmaps.

L89 – 16S rRNA gene sequencing (not 16S rDNA sequencing) is more appropriate to define the amplification of this marker gene. Please apply this also for the remaining text (e.g., L114). Also, check the standardization proposal by Marchesi and Ravel (2015).

Authors: terms were corrected through the whole manuscript

L93 – To better localize the reader, it would be good to add the country where the Vologda region is situated (Vologda region, Russian). Please, include it.

Authors: location was clarified

L97 – Seed sterilization is not possible because the endophytes are still inside. The term “disinfestation” is more appropriate. Please, change it.

Authors: the term was corrected

L99 – What does untreated soil meaon 3 formsn? Is it from the same region? Please clarify. Also, it would be easier for the reader to understand if a sentence is added highlighting that “the soil from every plant was mixed individually generating five unique soils. Then, every soil was seeded …).

Authors: yes, it’s a great suggestion. We rephrase the sentence

L122 – Please, change “beta-metrics” to “beta diversity”. It is the most appropriate term to be used. Also, include what was used to calculate the Bray-Curtis distance, the ASVs or any combined taxa level.

Authors: the term was corrected

L149 and L156 – Please, provide the table with all statistical results.

Authors: results are provided in supplementary

L136, L138, L153 – Bray-Curtis is misspelled. Please check the entire manuscript and figures, including legends.

Authors: misspelling is corrected

L137 – It is said that PCoA and NMDS could not clarify the diversity. It might be an effect caused by normalization (rarefaction). For alpha diversity, rarefaction is a common practice. But for beta diversity, the best approach would be using a square root normalization. When rarefying data, many species in low abundance are rarefied. They are important in the differences observed in the overall beta diversity. Please, look at McMurdie et al. 2014.

Authors: we use rarefied data for alpha-diversity only. The information about it now presented clearly

L146 – After Phase I, no need to add “one”.

 Authors: the sentence was corrected

L155 - Please, change “median distance” to “Bray-Curtis distance”.

Authors: the sentence was corrected

L177 – Observed is a metric to measure the richness of species. Please, change “Observed” to “richness”. Please apply this also for the remaining text.

Authors: the sentence was corrected except several cases, where we talk about exact Observed ASVs index (in particular, in variance analysis).

L216- L222 – Please check the text content here. It seems that has something wrong, or it doesn’t belong to your manuscript. Remove it or rewrite it in a meaningful way.

Authors: the text was rewritten

L224-L225 – Please, check your conclusion. It seems contradictory to what you wrote on L172-L173 or your Bray-Curtis results. 

Authors: the text was rewritten

FIGURE 1 – In Fig. 1A, change the Y axis to “Species richness” (instead of “Observed index”). Also, in the figure title, it should be “A. Richness” (instead of “A. Alpha-diversity). In Fig. 1B, the title should be “B. Bray-Curtis distance” (instead of Average …). In Fig. 1C, the title should be “C. Differential relative abundance”. What does the NA mean? Also, italic genera should be italicized. 

Authors: Differential relative abundance was removed completely. According fig 1A I suggest, that we measure exact Observed ASVs index (not richness in common), so, the Y-axis named after it. The title is changed to “Richness”

FIGURE 4 – This figure is not informative for your study. I suggest remove it from the manuscript.

Authors: we replaced the heatmap to PCoA plot

Round 2

Reviewer 2 Report

Authors have adequately addressed my concerns. 

Reviewer 3 Report

Dear Authors,

Thank you for the improvement in the manuscript.

Best regards.